# Spec-Driven Development: From Code to Contract in the Age of AI Coding Assistants

## Abstract

The rise of AI coding assistants has reignited interest in an old idea: what if specifications—not code—were the primary artifact of software development? Spec-driven development (SDD) inverts the traditional workflow by treating specifications as the source of truth and code as a generated or verified secondary artifact. This paper provides practitioners with a comprehensive guide to SDD, covering its principles, workflow patterns, and supporting tools. We present three levels of specification rigor—spec-first, spec-anchored, and spec-as-source—with clear guidance on when each applies. Through analysis of tools ranging from Behavior-Driven Development frameworks to modern AI-assisted toolkits like GitHub Spec Kit, we demonstrate how the spec-first philosophy maps to real implementations. We present case studies from API development, enterprise systems, and embedded software, illustrating how different domains apply SDD. We conclude with a decision framework helping practitioners determine when SDD provides value and when simpler approaches suffice.

## CCS Concepts

• **Software and its engineering** → **Software development methods**; **Specification languages**; *Software testing and debugging*.

## Keywords

Spec-Driven Development, AI-Assisted Coding, Behavior-Driven Development, Test-Driven Development, API Design First, Software Specifications

**ACM Reference Format:**
Anonymous Author(s). 2026. Spec-Driven Development: From Code to Contract in the Age of AI Coding Assistants. In *Proceedings of 3rd ACM International Conference on AI-powered Software (AIWare 2026)*. ACM, New York, NY, USA, 8 pages. https://doi.org/10.1145/nnnnnnn.nnnnnnn

## 1 Introduction

For decades, code has been the king of software development. Requirements documents exist, but they drift. Design diagrams are drawn, but they rot. Tests are written, but often after the fact. The code—whatever it actually does—becomes the de facto truth of the system.

This code-centric reality has consequences. When a new developer asks "what should this function do?" the answer is often "read the code." When a stakeholder asks "does the system meet our requirements?" the answer requires reverse-engineering intent from implementation. When an AI coding assistant is asked to add a feature, it must guess what the developer wants from a vague prompt.

Spec-driven development (SDD) [9, 27] offers an alternative: *make the specification the source of truth, and let code derive from it.* Instead of coding first and documenting later (or never), teams write clear specifications of intended behavior, then generate, implement, or verify code against those specifications. The spec becomes the authoritative description that both humans and machines use to understand, build, and maintain the system.

### 1.1 The AI Catalyst

While spec-first thinking is not new—Test-Driven Development (TDD) and Behavior-Driven Development (BDD) have advocated for it for years—the emergence of AI coding assistants [5, 10] has made SDD newly relevant. The problem is simple: AI models are excellent at pattern completion but poor at mind reading.

Consider a developer who prompts an AI with: "Add photo sharing to my app." The AI must guess: What format? What permissions model? What size limits? Cloud storage or local? Compression? The result is often plausible-looking code that makes dozens of unstated assumptions—many of them wrong. This is what practitioners call "vibe coding"—relying on loose prompts that lead to inconsistent or erroneous outputs from LLMs. By providing AI with unambiguous, executable contracts, SDD enhances the reliability of coding agents and opens new avenues for scalable software creation.

Now consider the same request with a specification: "Users can upload JPEG or PNG photos up to 10MB. Photos are stored in S3 with user-ID-prefixed keys. Only the uploader can delete their photos. Photos are resized to 1024px max dimension on upload." The AI now has enough information to generate code that matches intent.

> **Key Insight:** In spec-driven development, code is the implementation detail of the specification—not the other way around. The spec declares intent; the code realizes it.

### 1.2 What This Paper Provides

This paper serves as a practitioner's guide to spec-driven development. We begin by defining clear levels of specification rigor—spec-first, spec-anchored, and spec-as-source—and articulating when each approach applies. We then present a practical workflow for implementing SDD, examining how the approach works both with and without AI assistance. A survey of tools and frameworks follows, ranging from traditional BDD frameworks to modern AI-assisted toolkits. Case studies illustrate SDD in action across API development,

*AIWare 2026, Montreal, Canada*
2026. ACM ISBN 978-x-xxxx-xxxx-x/YYYY/MM
https://doi.org/10.1145/nnnnnnn.nnnnnnn

enterprise systems, and embedded software domains. Finally, we provide guidance on when SDD delivers value and when simpler approaches suffice.

## 2 The Specification Spectrum

Not all spec-driven approaches are equal. Teams adopt different levels of rigor depending on their needs, tooling, and domain constraints. Figure 1 illustrates the spectrum from traditional code-first development to fully spec-as-source approaches. Understanding where your team falls on this spectrum—and where it should be—is the first step in adopting SDD effectively.

### 2.1 Spec-First: Guided Initial Development

> **Definition: Spec-First**
>
> In **spec-first** development, a specification is written before coding to guide the initial implementation. Once code exists, the spec may or may not be maintained—the primary value is in the initial clarity it provides.

Spec-first represents the entry point to SDD. Before writing code, the developer or team articulates what the code should do, typically as a user story with acceptance criteria, a BDD scenario, or a detailed requirements document. The spec guides implementation, but once the code is written and tests pass, the spec may be discarded or allowed to drift.

The defining characteristic of spec-first development is that the specification is written before implementation begins, ensuring that developers have a clear target before they start coding. However, the code becomes the primary artifact post-implementation, and the spec may become outdated as the code evolves through subsequent iterations. This approach carries a lower maintenance burden than stronger specification disciplines, making it practical for teams that cannot commit to ongoing spec maintenance.

Spec-first works particularly well for initial feature development when working with AI coding assistants. The upfront spec prevents the AI from guessing at requirements, dramatically improving the quality of generated code. It is also valuable for prototypes and one-off features where the cost of maintaining a spec alongside code indefinitely is not justified. However, spec-first does not protect against drift over time—if the codebase will be maintained long-term, teams should consider spec-anchored approaches.

### 2.2 Spec-Anchored: Living Documentation

> **Definition: Spec-Anchored**
>
> In **spec-anchored** development, the specification is maintained alongside the code throughout the system's lifecycle. Changes to behavior require updating both the spec and the code, keeping them synchronized.

Spec-anchored development treats the spec as a living document that evolves with the codebase. When a feature changes, the spec is updated first or simultaneously with the code. Automated checks—typically in the form of tests derived from the spec—ensure that spec and code remain aligned. If they drift, tests fail, providing immediate feedback that the system's documentation no longer reflects its behavior.

In this approach, the specification and code evolve together as equal partners. Tests enforce the alignment between them, with BDD scenarios commonly serving as automated tests that run on every commit. The spec serves as always-up-to-date documentation that developers and stakeholders can trust. However, maintaining this alignment requires discipline and tooling support—teams must commit to updating specs whenever behavior changes.

Spec-anchored is the sweet spot for most production systems. It provides the benefits of clear documentation and verifiable requirements without demanding that code be fully generated from specifications. BDD frameworks like Cucumber exemplify this approach, enabling teams to write human-readable scenarios that execute as automated tests. For API development, OpenAPI specifications paired with contract testing tools like Specmatic achieve the same alignment between spec and implementation.

### 2.3 Spec-as-Source: Humans Edit Specs, Machines Generate Code

> **Definition: Spec-as-Source**
>
> In **spec-as-source** development, the specification is the only artifact humans edit directly. Code is entirely generated from the spec and should never be manually modified. Any change to behavior means changing the spec and regenerating.

Spec-as-source represents the most radical form of SDD. The specification becomes, in effect, the source code—just expressed at a higher level of abstraction. Developers think in terms of requirements and behavior; machines translate that into executable code. If you want to change functionality, you change the spec and regenerate—you never edit the generated code directly.

This approach, drawing on Design by Contract principles [18], fundamentally inverts the traditional relationship between specs and code: the specification is the primary artifact, and code is entirely derived from it. Manual code editing is either prohibited or confined to well-defined extension points. This requires mature, trusted generation tooling—developers must have confidence that generated code correctly implements the spec. In return, drift is eliminated by design: since code is regenerated rather than manually edited, spec and code are always aligned by construction.

Spec-as-source is already standard practice in domains with well-defined code generation, such as generating API server stubs from OpenAPI specifications, or producing certified embedded code from Simulink models. In the automotive industry, engineers routinely build control algorithms in Simulink, verify behavior at the model level through simulation, and generate certified C code that nobody hand-edits. Emerging AI tools like Tessl aim to extend this approach to general software development, representing a future where specifications truly become "the new source code." However,

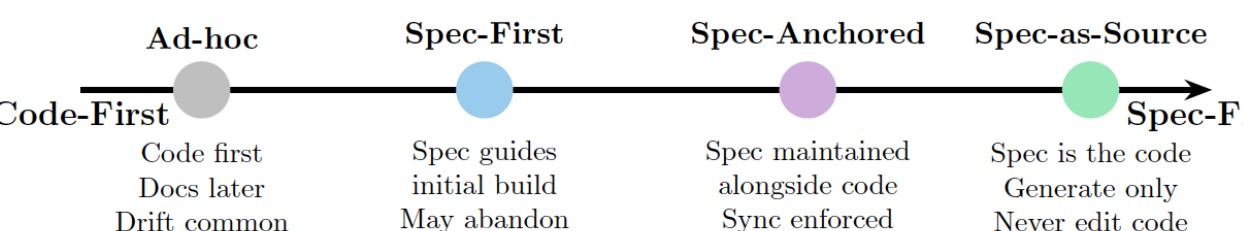

**Figure 1: The specification spectrum. Moving right increases the authority of specifications over code, but also increases the discipline required to maintain alignment.**

adopting spec-as-source requires high trust in generation quality and is currently practical only in domains where that trust has been established.

## 3 The SDD Workflow

How does spec-driven development work in practice? While specific tools vary, a common workflow emerges across SDD approaches. Figure 2 illustrates the four core phases. The key insight is that each phase produces an artifact that constrains and guides the next, creating a chain of accountability from intent to implementation.

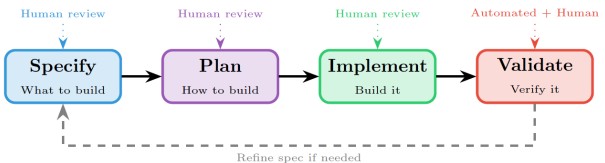

**Figure 2: The SDD workflow. Each phase produces an artifact that guides the next. Human review at each checkpoint ensures alignment with intent.**

### 3.1 Phase 1: Specify

The specify phase answers a fundamental question: *What should the software do?* The output is a functional specification describing behavior, requirements, and acceptance criteria—crucially, without prescribing implementation details. This separation of "what" from "how" is essential to SDD's power.

During this phase, teams articulate user-facing behavior through user stories, scenarios, and acceptance criteria. They define what success looks like using Given/When/Then format or input-output examples. Business rules and constraints are captured explicitly, and edge cases and error conditions

are identified upfront rather than discovered during implementation.

The quality of specifications directly determines the quality of everything that follows. Good specs share several characteristics: they are behavior-focused, describing what happens rather than how; they are testable, with each requirement being verifiable; they are unambiguous, meaning different readers reach the same interpretation; and they are complete enough to cover essential cases without over-specifying. Effective specs emphasize clarity, modularity, and self-checks that help guide AI agents during implementation.

> **Practitioner's Tip:** Write specs at the level of detail needed to remove ambiguity. If an AI or developer could interpret a requirement in multiple ways, add clarification. If there's only one reasonable interpretation, don't over-specify—excessive detail constrains implementation unnecessarily.

### 3.2 Phase 2: Plan

The plan phase answers a different question: *How should we build it?* Given the functional spec, this phase produces a technical plan covering architecture, data models, interfaces, and technology choices. Where the spec declares intent, the plan declares constraints on implementation.

Planning involves selecting technologies and frameworks appropriate to the problem, defining component architecture and boundaries, designing data models and schemas, specifying interfaces including APIs, messages, and contracts, and identifying non-functional requirements around performance, security, and scalability.

The plan phase bridges the "what" and the "how." It encodes constraints that the implementation must respect—for example, "use PostgreSQL for persistence" or "all API endpoints require authentication." When using AI coding assistants, the plan provides crucial context: the AI learns not just what to build but how the system is structured and what conventions it should follow. Without this context, even

a perfect functional spec may yield code that contradicts organizational standards or architectural decisions.

### 3.3 Phase 3: Implement

The implement phase produces working code that realizes the spec according to the plan. In traditional development, this is where most effort concentrates. In SDD, particularly with AI assistance, this phase may be substantially automated—but it still requires human oversight.

Implementation begins by breaking the plan into discrete, reviewable tasks. Each task is then implemented, whether by human developers, AI assistants, or a hybrid approach. Code is reviewed against both spec and plan to verify alignment. Unit tests are written to encode spec requirements as executable assertions.

A key SDD principle is working in small, validated increments. Rather than implementing the entire spec at once, teams break work into tasks where each delivers a testable piece of functionality. This enables frequent checkpoints where humans verify alignment, catching drift early before it compounds. Specifications act as "super-prompts" that break down complex problems into modular components aligned with agents' context windows, enabling AI systems to handle complexity that would overwhelm single-shot prompts.

### 3.4 Phase 4: Validate

The validate phase answers the crucial closing question: *Does the code actually meet the spec?* Validation closes the loop, ensuring that what was specified is what was built. This phase combines automated verification with human judgment.

Validation encompasses running automated tests at unit, integration, and acceptance levels, executing BDD scenarios against the implementation, reviewing for adherence to non-functional requirements, and conducting stakeholder acceptance testing where appropriate.

If validation reveals gaps—the code doesn't meet the spec— the team faces a decision: fix the code or revise the spec. If the original spec was wrong or incomplete, updating it is the right choice. If the code simply doesn't meet a valid spec, fixing the code is required. Either way, the spec remains the authority. This discipline ensures that specifications stay trustworthy—teams can rely on them because violations are detected and addressed, not ignored.

## 4 How SDD Boosts AI Coding Agents

Large language models like GPT-4 or Claude, when used as coding agents, benefit immensely from SDD by receiving optimized, context-rich inputs. Specifications act as super-prompts that break down complex problems into modular components aligned with agents' context windows. AI agents can generate code from specs while self-verifying against checklists for requirements adherence.

Empirical studies [14, 19], though nascent, suggest that human-refined specs significantly improve LLM-generated code quality, with controlled studies showing error reductions of up to 50%. This boosting effect is particularly evident in scalable scenarios: specifications enable parallel agent execution on non-overlapping tasks, with orchestration for dependencies. Teams can partition work at the spec level, allowing multiple AI agents to implement different components simultaneously without interference.

Challenges remain, including LLM non-determinism—even structured specs can lead to varying outputs. Techniques like property-based testing (PBT) address this by automatically verifying that invariants from specs are satisfied regardless of implementation variation. In embedded systems and other safety-critical domains, SDD combines LLM generation with formal verification to ensure compliance with standards like ISO 26262 [16]. Overall, SDD transforms AI agents from reactive tools into proactive collaborators, enhancing efficiency particularly in brownfield projects where legacy constraints are encoded in specifications.

An emerging approach involves "self-spec" methods where LLMs author their own specifications before generating code. The agent first produces a spec from a high-level prompt, which is then reviewed and refined by humans before the same or another agent implements against it. This creates an explicit separation between planning and execution, catching requirement misunderstandings before code is written.

## 5 Tools and Frameworks

A variety of tools support spec-driven development, from traditional testing frameworks to modern AI-assisted toolkits. Table 1 summarizes key categories. Common approaches include phased workflows (specify, plan, tasks, implement) and tools ranging from Kiro for VS Code-based specs to spec-kit for CLI-driven projects to Tessl for spec-as-source models.

**Table 1: Tools and Frameworks Supporting SDD**

| Category | Examples |
|---|---|
| BDD Frameworks | Cucumber, SpecFlow, Behave |
| TDD Frameworks | RSpec, JUnit, pytest |
| API Specification | OpenAPI, GraphQL SDL, Protobuf |
| Contract Testing | Pact, Specmatic |
| AI-Assisted SDD | Spec Kit, Kiro, Tessl |
| Model-Based | Simulink, SCADE |

### 5.1 Behavior-Driven Development (BDD) Frameworks

BDD frameworks allow teams to write specifications in near-natural language that can be executed as tests. The canonical format is Gherkin [7], which uses structured scenarios with Given/When/Then clauses (e.g., "Given the cart is empty / When I add item 'Widget' / Then the cart should contain 1 item"). These scenarios serve dual purposes: documentation that stakeholders can read and automated tests that

verify code. Tools like Cucumber [6], SpecFlow [23], and Behave [4] execute these scenarios against the application, bridging business requirements and technical implementation. The key insight: BDD scenarios are specifications, not just tests—write them before implementation and treat them as the authoritative description of feature behavior.

## 5.2 API Specification Tools

In API development, spec-driven approaches have been standard practice under the names "design-first" or "API-first" [24] for years. OpenAPI [21] (formerly Swagger) enables teams to define REST APIs with complete endpoint specifications, request/response schemas, and examples, then generate server stubs, client SDKs, and documentation from the spec. GraphQL SDL [13] allows teams to define types, queries, and mutations in a schema that becomes the contract between frontend and backend, enabling parallel development. For event-driven architectures, AsyncAPI [2] provides similar specification capabilities. Protocol Buffers [12] and gRPC [15] enable definition of service interfaces and message types with automatic generation of strongly-typed client and server code.

The benefit of API specification tools is clear: once the API spec is agreed upon, frontend and backend teams can work in parallel with confidence. The spec is the contract; any implementation that matches the spec is valid by definition. Contract testing tools like Pact [22] and Specmatic [25] automate verification that implementations actually match their specs.

## 5.3 AI-Assisted SDD Tools

Emerging tools structure AI coding workflows explicitly around specifications, recognizing that multi-step prompting with explicit artifacts yields better results than single-shot "just code this" prompts.

GitHub Spec Kit [11] is an open-source toolkit providing commands for spec-driven AI development. The workflow follows four explicit phases: `/specify` generates a detailed spec from a prompt, `/plan` creates technical architecture, `/tasks` breaks the plan into implementation tasks, and finally implementation generates code task by task. At each phase, humans review and refine before proceeding, maintaining alignment between intent and implementation.

Amazon Kiro [1] guides users through requirements, design, and task creation stages before any code generation begins. Kiro emphasizes structured requirements capture and iterative refinement, ensuring AI has clear context before attempting implementation. The explicit staging prevents the AI from guessing at requirements that were never specified.

Tessl [26] takes the most radical approach: spec-as-source where the specification is the maintained artifact and code is regenerated from it. Tessl represents the emerging vision of "specs as the new source code," where developers never edit generated code—they edit specs and regenerate.

These tools share a common insight: separating planning from implementation allows agents to focus on execution within defined boundaries, reducing the non-determinism that plagues loosely-prompted AI coding.

## 6 Case Studies

### 6.1 API-First Microservices

**Domain:** Financial services microservices — **Pattern:** Spec-anchored with OpenAPI — **Outcome:** 75% reduction in integration cycle time

A financial services company struggled with what they called "integration hell"—microservices frequently failed when deployed together because teams made incompatible assumptions about API contracts. Each team implemented their service in isolation, and incompatibilities only surfaced during integration testing, requiring expensive rework.

The company mandated API-first development as their solution. Before implementing any service, teams wrote OpenAPI specifications defining endpoints, request/response schemas, and error conditions. Consumer teams reviewed these specs and provided feedback before any coding began. This front-loaded the integration discussions that previously happened too late.

They used Specmatic [25] to generate mock servers from specs, allowing frontend development to proceed in parallel with backend work. More critically, Specmatic validated that implemented services matched their specs in CI. Any deviation caused the build to fail, preventing drift from accumulating.

Integration failures dropped dramatically after adoption. Teams reported a 75% reduction in cycle time for API changes because incompatibilities were caught at the spec review stage rather than in production. The specs became the contract that all parties trusted, eliminating the ambiguity that had caused so much rework.

### 6.2 BDD for Enterprise Features

**Domain:** Enterprise project management software — **Pattern:** Spec-anchored with Cucumber — **Outcome:** Stakeholder-verifiable requirements; reduced requirement ambiguity

An enterprise software team found that developers and product managers frequently disagreed on what "done" meant for features. Developers would implement something they believed met requirements, QA would find it didn't match product expectations, and arguments ensued about whose interpretation was correct. The lack of a shared, authoritative definition of expected behavior caused friction and rework.

The team adopted Cucumber for all user-facing features as their solution. Product managers wrote Gherkin scenarios describing expected behavior in plain language. Developers implemented step definitions to automate these scenarios as executable tests. A feature was only considered "done" when all its scenarios passed, providing an objective, verifiable definition of completion.

The Gherkin scenarios became a shared language that both business and technical stakeholders could read and

validate. Product managers could verify that scenarios captured their intent. When disputes arose, the scenario was the authority—if the scenario was wrong, it was updated with explicit agreement from stakeholders; if the code was wrong, developers fixed it. This eliminated the ambiguity that had caused so much rework and conflict.

## 6.3   Model-Based Embedded Development

**Domain:** Automotive engine control — **Pattern:** Spec-as-source with Simulink — **Outcome:** Verified control logic; certified code generation

An automotive supplier needed to develop engine control software that met ISO 26262 safety certification requirements. Manual coding was error-prone, and certification required tracing every line of code to specific requirements—a labor-intensive process when code was hand-written.

The team used MathWorks Simulink [17] to model control algorithms as block diagrams with state machines. The model was the specification: engineers simulated and verified behavior at the model level, catching algorithmic errors before any code existed. Once the model was verified through simulation, code was auto-generated from the model using a certified code generator.

The model-to-code generation was itself certified, meaning the generated C code was guaranteed to behave as the model specified. Engineers never edited generated code; if control logic needed changing, they changed the model and regenerated. This ensured the verified model and deployed code remained perfectly aligned by construction.

This approach embodies spec-as-source at its most rigorous: the specification (Simulink model) is the only artifact humans modify, and the implementation (C code) is entirely generated. SDD in embedded systems combines LLM generation with formal verification to ensure safety-critical compliance, demonstrating how specs ensure precision in domains like automotive and aerospace where errors can be catastrophic.

## 7   When to Use SDD

Spec-driven development is not universally applicable. Like any practice, it has costs—upfront spec effort, tooling investment, and discipline requirements—and benefits—clarity, quality, and maintainability. The decision framework in Figure 3 helps practitioners determine when SDD adds value.

SDD adds clear value when using AI coding assistants, as specifications dramatically improve output quality by removing the ambiguity that forces AI to guess. Complex requirements benefit from SDD because stakeholders can validate that the system meets their needs before code is written. Systems with multiple maintainers benefit because specs serve as documentation that survives team turnover. Integration-heavy systems gain from API specs that enable parallel development and prevent integration failures. Regulated domains often mandate traceability from requirements to implementation, which SDD provides naturally. Legacy modernization efforts benefit because extracting a spec from

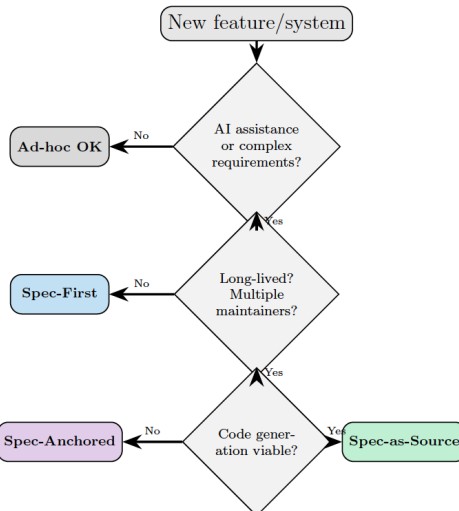

**Figure 3: Decision framework for selecting SDD approach. Start with the level of rigor that matches your needs.**

existing behavior enables clean reimplementation with confidence.

However, SDD may be overkill in certain situations. Throwaway prototypes don't justify spec investment that will be discarded. Solo, short-lived projects may find the overhead exceeds benefits when there's only one developer and no long-term maintenance. Exploratory coding suffers from premature specification that constrains learning when you don't yet know what you're building. Simple CRUD applications with obvious requirements need minimal spec—if requirements are unlikely to be misinterpreted, elaborate specifications add cost without value.

## 8   Common Pitfalls

Teams adopting SDD often encounter predictable challenges that can undermine the approach's benefits if not addressed.

**Over-specification** occurs when teams write specs that are too detailed, essentially becoming pseudo-code. This defeats the purpose of SDD, which is to separate "what" from "how." If your spec reads like code, you've gone too far—you've constrained implementation unnecessarily and lost the abstraction benefit that makes specs valuable.

**Specification rot** affects spec-anchored approaches when teams fail to update specs as code changes. The spec drifts from reality, losing its value as documentation and eroding trust. The solution is automated enforcement through tests that fail when spec and code diverge, making drift visible and painful rather than silent and accumulating.

**Specification as bureaucracy** emerges when specs become forms to fill out rather than tools for clarity. If the

specification process adds overhead without improving understanding or quality, teams will game the system or abandon it. Specs should be the minimum needed to remove ambiguity, not comprehensive documentation exercises.

**Tooling complexity** can overwhelm teams, particularly with AI-assisted tools that generate elaborate artifacts. Teams may drown in generated plans, task lists, and intermediate documents. The solution is to start simple and add tooling complexity only when it demonstrably helps—avoid cargo-culting elaborate workflows that add process without value.

**False confidence** is perhaps the subtlest pitfall. A passing spec test doesn't guarantee correct software—it only guarantees that the software matches the spec. If the spec is wrong, the code will faithfully implement the wrong thing. Specifications require the same careful review as code; they are not a silver bullet that eliminates the need for human judgment about requirements.

## 9 SDD vs Traditional Design Documents

A natural question arises: how is SDD different from traditional High-Level Design (HLD) and Low-Level Design (LLD) documents that software engineering has always used? After all, HLD describes architecture, LLD details implementation, and requirements documents specify functionality. Aren't these already specifications?

The answer is nuanced. Traditional design documents are specifications—the difference lies not in what is written, but in how it is used and whether it stays aligned with code. Traditional software engineering produces many specification-like artifacts: Software Requirements Specifications (SRS) for functional and non-functional requirements, High-Level Design documents for architecture and components, Low-Level Design documents for class diagrams and algorithms, and interface specifications for API contracts and IDL.

The problem is not the absence of specs—it's that they drift. By Sprint 3, the HLD is outdated. By release 2, the SRS no longer matches the product. The code becomes the de facto truth, and the documents become historical artifacts that nobody trusts or updates.

> **Key Insight:** Traditional design documents are *advisory*—developers read them, then write code that hopefully matches. SDD specs are *enforced*—tests fail if code diverges, and in spec-as-source approaches, code is regenerated rather than manually edited.

What SDD actually adds is threefold. First, executable specifications: traditional specs are read by humans, while SDD specs are executed as BDD scenarios, API contract tests, or model simulations—if code doesn't match, the build fails. Second, CI/CD integration: modern SDD embeds spec validation into continuous integration, checking every commit against the spec so drift is caught immediately rather than during quarterly reviews. Third, AI consumption: traditional design docs were written for human readers, while SDD specs are structured so AI coding assistants can consume them, generating code and tests from specs rather than guessing from vague prompts.

SDD is not a revolution—it's an evolution. The core insight (write specs first, let code derive from them) has been agile wisdom for decades. What's new is better tooling that makes executable specs practical, CI/CD maturity that enables automated enforcement, and AI as consumer where spec quality directly determines output quality. As Bryan Finster observed [8]: "SDD is not a revolution... it's just BDD with branding." But the branding serves a purpose: it reminds practitioners that specs should be authoritative, not advisory, and that modern tooling can enforce what was previously left to human discipline.

## 10 Relationship to Existing Practices

SDD is not a replacement for existing development practices—it builds on and extends them in the context of AI-assisted development.

Test-Driven Development (TDD) [3] is SDD at the unit level. Writing a test first is writing a micro-specification that defines expected behavior before implementation. SDD extends this thinking to higher levels—features, systems, and architectures—applying the same "specify first" discipline at broader scope.

Behavior-Driven Development (BDD) [20] is the most direct ancestor of modern SDD. Gherkin scenarios are executable specifications that bridge business requirements and technical implementation. What AI-assisted SDD tools add is assistance in generating code from those specs, accelerating the path from scenario to working software.

Domain-Driven Design (DDD) aligns well with SDD through its emphasis on ubiquitous language—specs written in domain terms that both developers and stakeholders understand. The shared vocabulary that DDD advocates becomes the foundation for specifications that are meaningful to all parties.

Agile methodologies are compatible with SDD. User stories with acceptance criteria are specifications; the Definition of Done is a form of spec. The difference is emphasis: SDD treats these artifacts as authoritative rather than advisory, and enforces alignment through automation rather than relying on human discipline alone.

## 11 Conclusion

Spec-driven development inverts the traditional relationship between specifications and code. Instead of code being the source of truth with documentation as an afterthought, SDD makes specifications authoritative and code derivative. This inversion becomes increasingly important as AI coding assistants become more capable—when an AI can generate code from specifications faster than humans can type, the bottleneck shifts to specification quality.

Specifications remove ambiguity for both human developers and AI assistants, preventing the guesswork and misinterpretation that leads to costly rework. The three levels of rigor—spec-first, spec-anchored, and spec-as-source—provide options that match different project needs, from lightweight initial clarity to rigorous code generation. Mature tooling

exists across the spectrum, from BDD frameworks and API specification tools to AI-assisted SDD toolkits, making spec-first workflows practical today. Teams should match rigor to need, using the minimum specification discipline that removes ambiguity for their context rather than over-engineering process.

SDD builds on decades of TDD and BDD wisdom while adapting these practices for the AI era. The ideas are not new; what's new is the tooling and AI capabilities that make specs more powerful than ever. Work is being redefined as developers shift from manual coding to orchestrating specifications, reviewing AI outputs, and focusing on high-level design.

As software systems grow more complex and AI becomes more capable, the question shifts from "what code should I write?" to "what specification should I provide?" Teams that master spec-driven development will get more value from AI tools while maintaining the clarity and traceability that complex systems require. SDD offers a framework for answering that question systematically—making specifications, not code, the primary artifact of software development.

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
