# OpenReview forum: "Spec-Driven Development: From Code to Contract in the Age of AI Coding Assistants"
_ACM.org/AIWare/2026/Conference — Submitted to AIware 2026_

### Official Review · Reviewer_NxxP · 2026-02-26

**Rating:** 2
**Confidence:** 3

**Review:**

### Strengths
1. The paper closely relates to the real-world needs of AI-assisted development, presenting a novel and practically appealing topic.
2. It provides a framework for achieving specific goals, offering a direct reference point for engineering teams to introduce SDD.

### Weaknesses
1. Insufficient rigor: Claims lack reliable peer-reviewed literature.
2. Lack of necessary evidence: The case study in Section 6 lacks a clear research object, task definition, evaluation metrics, and analytical methods, making it difficult to assess its reliability and generalizability.
3. Insufficient details: Some content remains at the conceptual level. The decision-making framework lacks measurable metrics and implementation details, making it difficult to guide readers in evaluating and implementing the proposed technology in real projects.
4. Readability issues: The lack of an introduction to the background/related work and terminology makes it unfriendly to non-practitioner readers and weakens the paper's overall readability.

### Comments
1. The viewpoints (e.g., the key insight in the Introduction) are not novel. SDD is an existing field (existing literature cited in Line 69), and in AI assistant coding, specification-driven LLM code generation methods have already received widespread attention in the SE field [1][2]. I suggest that the authors could emphasize the differences and incremental contributions between this paper and related work in Section 1.2.
2. The paper cites too many no peer-reviewed papers, mostly blog posts or grey literature, with only a few preprints. I respect the potential forward-looking nature of this work. However, such a citation list is insufficient to support a rigorous academic paper.
   1. For example, the "Empirical study" mentioned in Line 404 is actually two online blog posts.
   2. There are no relevant citations when TDD and BDD are first introduced in Line 80.
3. The case study in Section 6 lacks methodological information, making it impossible to judge its credibility and reproducibility. Specifically, readers cannot determine from the current description whether the cases are real projects or example scenarios constructed by the authors. Furthermore, the case study lacks detailed methodology and statistical analysis (e.g., research subjects, task definition, evaluation metrics, etc.). The paper should clarify this type of information and supplement or cite possible supporting materials (if possible), which can significantly improve the reliability of the claim that "SDD is effective/feasible in AI coding scenarios" in this paper.
4. Many parts of this paper remain at the conceptual level, lacking actionable details and measurable metrics. For example, Figure 3 provides a decision framework for using SDD. If the authors could propose more specific metrics or provide some inspiration for design metrics (e.g., what metric can be used to evaluate the complexity of requirements? What can be used to evaluate code generation viability?), the technical value and contribution of the paper would be further enhanced.
5. Minor
   1. I appreciate the presentation style of the first two paragraphs of the Introduction. However, as a rigorous academic paper, are there any actual literature documents to support these claims or observations? (e.g., Lines 48-50, Lines 52-66) Please provide citations.
   2. There is a problem with the font formatting; it is inconsistent with the font type in the ACM proceeding template.
   3. Some abbreviations do not provide full names or references, such as `CRUD` in Line 673.
   4. In addition, this paper does not provide a section (background or related work) to introduce the related work, necessary knowledge, or terms, which reduces the readability for non-practitioner readers.

[1] SpecGen: Automated Generation of Formal Program Specification Generation based on LLMs. ICSE 2025

[2] Self-Spec: Model-Authored Specifications for Reliable LLM Code Generation. Agents4Science 2025

**Summary:**

This paper focuses on the practical value of Spec-Driven Development (SDD) in the era of AI-assisted coding, advocating for **specifications** rather than code to become the primary artifacts, and providing a guide for practitioners.
Overall, this topic is novel and interesting: using specifications to constrain and organize AI-assisted programming does indeed align with current popular topics in SE practice.
However, the paper currently reads more like a viewpoint/popular science article for practitioners: its key arguments lack reliable peer-reviewed literature support, and case studies and conclusions lack verifiable methodological details and chains of evidence, resulting in insufficient technical rigor, reproducibility, and operability.

---

### Official Review · Reviewer_umxm · 2026-03-04

**Rating:** 2
**Confidence:** 5

**Review:**

The paper addresses a highly timely and relevant problem, namely how software engineering practices should evolve in the era of AI-assisted development and coding agents. The central argument—that well-structured specifications can serve as effective inputs (“super-prompts”) for AI coding assistants—is compelling and aligns well with current trends in agent-based software development. The paper provides a clear conceptual framework by organizing spec-driven development into three levels of rigor and presenting a well-structured workflow that connects requirements, architectural planning, implementation, and validation. The presentation is very clear and accessible, making the ideas easy to understand for both academic and practitioner audiences. Additionally, the paper situates SDD within existing practices such as TDD, BDD, and API-first development, which helps readers connect the proposed perspective to established software engineering methodologies. The discussion of tools and real-world scenarios further strengthens the practical relevance of the work.

However, Despite its clarity and relevance, the paper’s research novelty and empirical grounding are limited. Much of the proposed concept builds upon existing practices such as TDD, BDD, contract-based development, and model-driven engineering, and the paper does not clearly articulate what fundamentally new research contribution SDD introduces beyond reframing these approaches in the context of AI coding assistants. To address this, the authors could better emphasize novel research questions or mechanisms, for example by formalizing the role of specifications in AI-assisted workflows or by defining measurable properties of specification quality for AI-generated code.

A second limitation is the lack of empirical evaluation. The paper relies primarily on conceptual discussion and illustrative case studies, which remain largely anecdotal and do not provide quantitative evidence of the benefits of SDD. This could be improved by conducting controlled experiments or empirical studies, such as evaluating the effect of structured specifications on the quality, correctness, or maintainability of AI-generated code.

Finally, while the paper discusses various tools that support spec-driven workflows, it does not introduce a new method, system, or artifact. Providing a prototype tool, workflow implementation, or experimental framework demonstrating how SDD concretely improves AI-assisted development would significantly strengthen the technical contribution of the work. The work currently reads more as a vision or position paper rather than a fully developed research contribution.

**Summary:**

This paper introduces Spec-Driven Development (SDD) as a development paradigm in which specifications become the primary artifact of software development while code is treated as a derived implementation artifact. The paper argues that the emergence of AI coding assistants and agentic software development workflows makes specification-centered development increasingly important. It proposes a conceptual framework describing three levels of specification rigor—spec-first, spec-anchored, and spec-as-source—and outlines a structured workflow consisting of specification, planning, implementation, and validation phases. The paper also surveys existing tools and practices supporting SDD and illustrates the approach through several industry-inspired case studies, including API-first microservices, behavior-driven development in enterprise systems, and model-based development in safety-critical domains. Overall, the paper aims to position SDD as a practical methodology for improving the reliability and effectiveness of AI-assisted software development.

---

### Official Review · Reviewer_NuGa · 2026-03-11

**Rating:** 2
**Confidence:** 4

**Review:**

I have the following concerns regarding this submission:
- The paper does not present a clear research goal or research questions, nor does it describe a well-defined research methodology or study design.
- The paper does not provide empirical evidence to support the claims made.
- Many of the claims are not supported by peer-reviewed academic references. Instead, most of the cited sources are non-academic, such as blog posts.
- The manuscript does not follow the typical structure of an academic paper (e.g., Methodology, Results, and Discussions).
- A replication package is not provided for the study.

**Summary:**

The paper introduces Spec-Driven Development (SDD) as an approach that treats specifications as the primary source of truth in software development. The authors argue that as AI coding assistants become more prevalent, clear specifications are essential because AI models need unambiguous requirements rather than vague prompts. The paper presents three levels of specification rigor: spec-first (write specs before coding), spec-anchored (maintain specs with automated verification), and spec-as-source (generate code from specs). It outlines a four-phase workflow including specify, plan, implement, validate and lists supporting tools from traditional BDD frameworks to AI-assisted toolkits. The authors conclude that as AI capabilities grow, the developer's role shifts from manual coding to crafting specifications and orchestrating AI agents.